# Side effects may include: Consequence neglect in generating solutions

Christopher Rodriguez*, Daniel M. Oppenheimer

Department of Social & Decision Sciences, Carnegie Mellon University, Pittsburgh, Pennsylvania, United States of America,

* crodrig3@andrew.cmu.edu

## Abstract

Strategies designed to address specific problems often give rise to unintended, negative consequences that, while foreseeable, are overlooked during strategy formulation and evaluation. We propose that this oversight is not due to a lack of knowledge but rather a cognitive bias rooted in focalism—the tendency to focus narrowly on the primary objective, ignoring other relevant factors, such as potential consequences. We introduce the concept of consequence neglect, where problem solvers fail to generate or consider downstream effects of their solutions because these consequences are not central to the proximal goal. Across four studies, we provide evidence supporting this phenomenon. Specifically, we find that individuals rate strategies more negatively after being prompted to generate both positive and negative consequences, suggesting that negative outcomes are not naturally weighted unless attention is explicitly drawn to them. We conclude by discussing the broader implications of consequence neglect for policymaking, business, and more general problem solving, and offer directions for future research.

## Introduction

"There was an old lady who swallowed a fly; I don't know why she swallowed a fly - Perhaps she'll die!

There was an old lady who swallowed a spider, that wriggled and jiggled and tickled inside her! She swallowed the spider to catch the fly; I don't know why she swallowed a fly - Perhaps she'll die!

There was an old lady who swallowed a bird; How absurd to swallow a bird!..."

- Rose Bonne

In a classic children's song, an old lady swallows progressively larger animals, each intended to solve the issue of having swallowed the previous one, but ultimately

**Data availability statement:** The data within our manuscript can be found in the files tab of the following OSF link: https://osf.io/8jrew/

**Funding:** The author(s) received no specific funding for this work.

**Competing interests:** The authors have declared that no competing interests exist.

leading to bigger problems down the road. The rhyme concludes: "There was an old lady who swallowed a horse; she died of course!" While silly and absurd, the song contains an important moral for children: the importance of considering the future consequences of their actions. In every verse it was clear that swallowing a larger animal would have negative outcomes, but the old lady repeatedly failed to anticipate them. Of course, the old lady of the song is fictional, but her consequence neglect mirrors a common pattern in real-life decision making where obvious future consequences of current actions are often ignored.

For example, during the British occupation of India, to combat the growth of the cobra population, the British created a bounty for cobra skins. While the goal was to encourage people to hunt cobras, thus reducing the cobra population, it overlooked an obvious consequence: cobra breeding (with the purpose of selling their skin) developed into a lucrative business. The British were then stuck paying for many cobra skins without meaningfully reducing the cobra population. To resolve this, the British called off the bounty. Unfortunately, this resulted in another foreseeable side effect: when cobra farming stopped being profitable, cobra farmers simply releasing their livestock into the wild. Suddenly, the British found themselves with a cobra problem more severe than before [1].

This fiasco could have been avoided if only the British had thought through what the likely consequences of various policies (above and beyond those intended by the policies) might be. Such oversights are common. This paper aims to address why. We suggest that when there is a problem that needs to be solved, evaluators tend to have "tunnel vision" that focuses on how a given solution will solve that problem (achieve the proximal goal) to the exclusion of considering other consequences that the solution might engender. That is to say, the problem-solving process leads people to consider too heavily the question of "how well does this solution solve the problem?" at the expense of the question "how good is this solution wholistically?" Consideration of potential downstream consequences is crucial for evaluating a solution wholistically. We suggest this is important in both formal policy-making contexts (e.g., imposing tariffs) as well as everyday problem-solving domains that all people may face (e.g., motivating a teen to study). Thus, we say that someone is exhibiting *consequence neglect* when they fail to consider consequences that materially affect their evaluation of a policy (or a problem-solution more generally), despite being capable of predicting those consequences.

We are not the only scholars who have observed obvious consequences going overlooked. For instance, Bazerman [2] discusses climate change as a "predictable surprises" which he defines as "an event or set of events that catch an organization off-guard, despite leaders' prior awareness of all of the information necessary to anticipate the events and their consequences." He argues that scientists have long been telling us about the dire effects of climate change, and thus, disasters driven by the effects of climate change are predictable surprises. In previous work, Watkins & Bazerman [3], outline a framework which can be used to describe predictable surprises and when they may occur. Specifically, their framework suggests something to be a predictable surprise if it falls into one of three categories: 1) the emerging

problem was not recognized and should have been, 2) the emerging problem was recognized but not prioritized when it should have been, or 3) the emerging problem was recognized and prioritized but a response was not mobilized when it should have been. To utilize this framework, we believe the phenomenon of consequence neglect lives within category 1, when the emerging problem was not recognized but should have been. The British would not have thought the cobra bounty policy to be an effective policy if they had considered the predictable surprise, (or in our terms, non-focal consequence) of people starting cobra farms.

Other scholarly work related to consequence neglect lies in the phenomenon of *opportunity cost neglect* [4], defined as when consumers fail to consider alternative uses for their money when making a consumption decision. The authors show that prompting consideration of opportunity costs leads to a reduction in purchase rates. One could construe opportunity cost neglect as a form of consequence neglect. When considering a purchase, consumers focus on whether they like the product, while neglecting the consequence that if they spend their money on the product under consideration, they will not have that money in the future to spend on other, possibly more desirable products. More recent literature has extended this finding from consumer behavior to public policy, showing that participants taking on the role of policy makers were less likely to invest in specific policies when reminded about the opportunity costs of doing so [5].

Why would people neglect consequences when doing so can lead to suboptimal decisions? Watkins & Bazerman [3] highlight self-serving biases, and a number of, organizational, and political reasons for why predictable surprises may occur. We do not dispute the importance of such factors in creating predictable surprises, but here we work at a different level of analysis, showing how basic cognitive mechanisms can lead people to neglect consequences. It is well known that people have limited cognitive capacity [6] and shown that people take cognitive shortcuts to reduce the mental burden of difficult problems [7]. One common way in which people simplify decisions is by reducing the amount of evidence that they consider (for a review see [8]). Previous research has documented considerable evidence that when people are focused on particular goals, cues, or features, they often neglect other relevant information that would increase the quality of their decision making: a phenomenon known as *focalism*. For instance, Wilson et al [9] asked football fans how their happiness would be influenced by the outcome of a football game of a team they supported. The question focused participants on a specific source of happiness (the happiness caused by their favored team winning or losing) leading participants to overestimate the extent to which their overall happiness would be impacted by the game and neglect how other events in their lives (e.g., upcoming holidays, romantic interactions, work obligations, etc.) would affect their happiness. However, when participants' attention was drawn to the fact that those other future events would also impact their happiness, participants ceased to overweight the effect of the game. Schkade and Kahneman [10] document a similar focusing illusion in the effect of weather on predictions of life satisfaction. Focalism even influences how academics design studies and interpret research results [11].

We argue that even when people are capable of generating unintended consequences of a solution to a given problem, they may not attempt to do so because these consequences are not naturally focal to the development of the solution. In problem solving contexts, most focal to the evaluation is the solution's effectiveness at dealing with the primary problem at hand. However, many consequences are not directly related to the initial problem, even though they are crucial to a proper evaluation of the solution as a whole; we call such consequences *material consequences*. For instance, the old woman in the opening example was focused on how to catch a fly in her stomach, and the British were focused on how to get people to kill cobras, those are focal consequences. However, the fact that the old woman would now have a spider in her stomach, and the British had incentivized breeding cobras are material but non-focal consequences of their solutions; they are the direct results of their solutions, but are unrelated to the initial problem to be solved. Due to focalism, material but non-focal consequences of a solution may never be called to mind, leading to the adoption of less effective (or even harmful) solutions. Thus, asking a consequence neglectful person to explicitly consider consequences should lead to better policy outcomes because it forces them to confront material consequences that were not focal to the policy's initial goal.

Other scholars have documented domains in which changing what is focal in our decision context can lead to shifting preferences [12,13]. Thus, providing a decision-making framework that calls for explicit consideration of consequences could serve as a relatively minor nudge [14] with a potential meaningful impact on how policies are selected and chosen to be implemented.

In order to explore consequence neglect, we use an experimental paradigm stemming from the illusion of explanatory depth literature (IOED) [15]. We use this paradigm not because we think that the theoretical construct of consequence neglect is related to the illusion of explanatory depth, but instead because it allows a convenient within-subjects method which can be used to detect evidence for consequence neglect. In classic studies, participants evaluated their understanding of the functional mechanisms of a variety of objects (e.g., toilets or helicopters) both before and after attempting to give a detailed explanation of the mechanism. This allowed the authors a within-subject method of measuring how much people overestimated their explanatory prowess. Notably, when evaluating their knowledge of how objects work, participants tend to focus on the function or purpose of the object (which they do understand), rather than on the mechanisms that allow the objects to achieve their function/purpose (which they do not understand) [16]. This leads people to overestimate their understanding unless their focus is drawn to the steps of the mechanistic process.

Our studies utilize a similar structure. In each of our studies, participants were presented with predicaments. Participants then either read a proposed solution or generated their own solution (depending on the study) and evaluated how favorable they found the proposed solution. Subsequently, participants generated (in a free response format) two positive and two negative consequences of the proposed solution. Finally, participants again evaluated how favorable they believed the proposed solution to be.

If participants had considered consequences in their initial evaluation, then highlighting the consequences should have little impact on their subsequent re-evaluation; those consequences would already have been incorporated into their judgments of favorability. However, if despite being capable of generating consequences material to a policy's evaluation, participants initially neglected consequences, then making consequences focal (by explicitly asking participants to consider them) should lead participants to updates to their assessments of the policy's quality.

To foreshadow the results, across multiple policy contexts, participants reliably evaluate policies to be of lower quality after having been prompted to generate both positive and negative consequences for those policies. This pattern of results is consistent with the idea that future benefits (solving the proximal problem) had already been incorporated into evaluations, but future costs (unintended negative consequences) had not. It is notable that in several of these studies participants generated these consequences themselves; thus, they could have been taken into consideration during their initial evaluation (i.e., these aren't consequences that were unforeseeable). Below, we summarize each study and its key findings.

## Study overview

### Study 1: Establishing the existence of consequence neglect.

In our first experiment, participants were presented with six pre-selected solutions for addressing different societal and organizational problems. They rated the effectiveness of these policies before and after being prompted to generate both positive and negative consequences. We observed a consistent downward shift in policy ratings after the consequence generation task, suggesting that negative consequences were not initially considered. Additionally, we explored whether individual differences (e.g., cognitive reflection, demographic variables) predicted consequence neglect but found no significant associations.

### Study 2a: Consequence neglect in self-generated solutions.

Building on Study 1, Study 2a examined whether consequence neglect extends to self-generated solutions. As in Study 1, a consequence generation task led to significant downward revisions in ratings, confirming that individuals neglect

negative consequences even in solutions they personally devise. This finding suggests that consequence neglect occurs not only in policy evaluation but also during the problem-solving process itself.

### Study 2b: Comparing self-generated vs. other-generated policies.

Study 2b tested whether consequence neglect differs between solution generators and solution evaluators. To allow direct comparison, we took solutions generated in Study 2a and presented them to a new group of participants for evaluation. The consequence generation task once again led to significant downward revisions in policy ratings. Notably, the magnitude of consequence neglect was statistically similar for self-generated and other-generated solutions, suggesting that the failure to consider negative consequences is equally likely to occur during both policy creation and evaluation.

### Study 3: Ruling out alternative explanations.

To ensure that our results were driven by consequence neglect rather than unrelated cognitive processes, such as fatigue or mere re-evaluation of the policy. Study 3 introduced two control conditions: a thought generation condition (where participants reflected on policy implementation rather than consequences) and a re-evaluation only condition (where participants simply rated policies twice without any intermediate task). Only the consequence generation condition led to significant downward rating shifts, ruling out the possibility that mere re-evaluation or general cognitive engagement accounted for the effects observed in Studies 1 and 2.

## Study 1: Consequence neglect proof of concept

In our initial study, we set out to determine if consequence neglect was, in fact, an observable phenomenon. In addition, we wanted to explore possible individual differences that might be associated with consequence neglect. As such, we developed a scale meant to capture the degree to which someone has a propensity for consequence neglect (a rating of individual differences in consequence neglectfulness). In addition, we measured people's scores on Frederick's [17] Cognitive Reflection Task (CRT), a measure of proclivity for deliberative thought. If intuitive thought leads to evaluation based upon how well it solves the problem, and deliberative thought is necessary to evaluate policies more holistically (i.e., considering consequences), then we would expect lower scorers on the CRT to be associated with higher rates of consequence neglect.

### Method

*Participants.* One hundred participants were recruited from Amazon Mechanical Turk for a flat rate payment of $3.90 based on the estimated time to complete the survey. We took two steps to ensure data quality: 1. Upfront attention checks were used to screen out potential bots or inattentive subjects. 2. Before exiting the survey, participants were given an open-ended free response question that sought to differentiate humans from bots [18]. After accounting for these screens, our final sample size for analysis consisted of 95 participants (36% female, mean age = 40 years). The Carnegie Mellon University Institutional Review Board approved all aspects of this study as well as all subsequent studies prior to data collection. Upon entry into the study, informed consent was received from participants via survey response.

 *Procedure.* After completing the aforementioned screening procedures and agreeing to participate in the study, participants were presented with a problem and a policy aimed at solving that problem. The participants evaluated the likely effectiveness of the policy on 5-point Likert scales (1=Not effective at all, 5= Extremely effective). This process was repeated for six different problems/policies that included: developing an incentive scheme to increase learning in schools, motivating a salesforce, improving quality of life in a low-income neighborhood, preventing negative outcomes of cosmetic surgery, raising the minimum wage, and responding as a CEO to a company controversy. For example:

"Imagine that you are a policy maker considering how to improve the learning in our education system. The currently proposed policy is to offer incentive bonus pay to teachers for higher student test scores. How effective do you think this policy would be?"

The complete wording for all of the stimuli can be found in our Web Appendix (which is in the files tab of the following OSF link: https://osf.io/8jrew/).

Participants were also presented with another problem concerning the prevention of office supply theft. However, for this problem they were asked to generate their own solution (rather than being provided with an experimenter generated solution). This prompt read:

"Imagine that someone in your office has been stealing supplies (i.e., pens and paper). Your boss has put you in charge of finding a solution to prevent this theft from occurring. Your boss likes to provide these supplies because she thinks it boosts productivity, so ceasing to offer supplies is not an option. What solution would you propose to your boss to solve this problem?"

After generating their solution, they rated the solution's effectiveness on the same scale as the previous ratings. Next, participants completed a demographics questionnaire in which they self-reported age, education level, race, gender, income, and political affiliation. Participants also completed the CRT [17] as well as a novel individual difference scale meant to measure one's consequence neglectfulness. The novel scale consisted of 11 items that participants responded to on 5-point agreement scales (1=strongly disagree, 5=strongly agree). Example items from the scale include (complete scale can be found in our Web Appendix which is in the files tab of the following OSF link: https://osf.io/8jrew/)

"I often ask myself 'how did I not see this coming?'"

"When brainstorming, once I find a solution that seems to solve the problem, I choose it and don't worry about what the long-term consequences may be."

Participants were next presented with the consequence generation task. For each of the six policies that they had previously seen, as well as their own solution to the office supplies problem, participants were asked to generate two positive and two negative consequences. The order in which positive and negative consequences were generated was counterbalanced to control for possible order effects.

After participants completed the consequence generation task for all the policies, participants were again presented with all seven of the policies and asked to rate each of them using the same scale as on their initial ratings. After this final rating elicitation, participants completed the outro bot-check and were debriefed. A visual representation of the experimental paradigm can be found in Fig 1.

*Consequence neglect*. Our design allows us to test for evidence supporting consequence neglect within subject. Prior to the first rating, participants may or may not have called particular consequences of a given policy to mind. If these consequences were thought of at the time of policy evaluation, then they would be reflected in the initial round of policy ratings. However, if participants did not initially call these consequences to mind during the first evaluation, then the initial ratings would not reflect the consideration of these consequences. During the second rating, we know explicitly that participants had called consequences to mind since we required them to do so. Thus, if we find a difference between the first and second rating, we can attribute it to engagement in the consequence generation task. It is notable that since participants are responsible for generating the consequences themselves, we know that they were capable of coming up with these consequences. Thus, a systematic downward shift in ratings would be consistent with the presence of negative consequence neglect because people were perfectly capable of generating these consequences substantive to policy evaluation but did not.

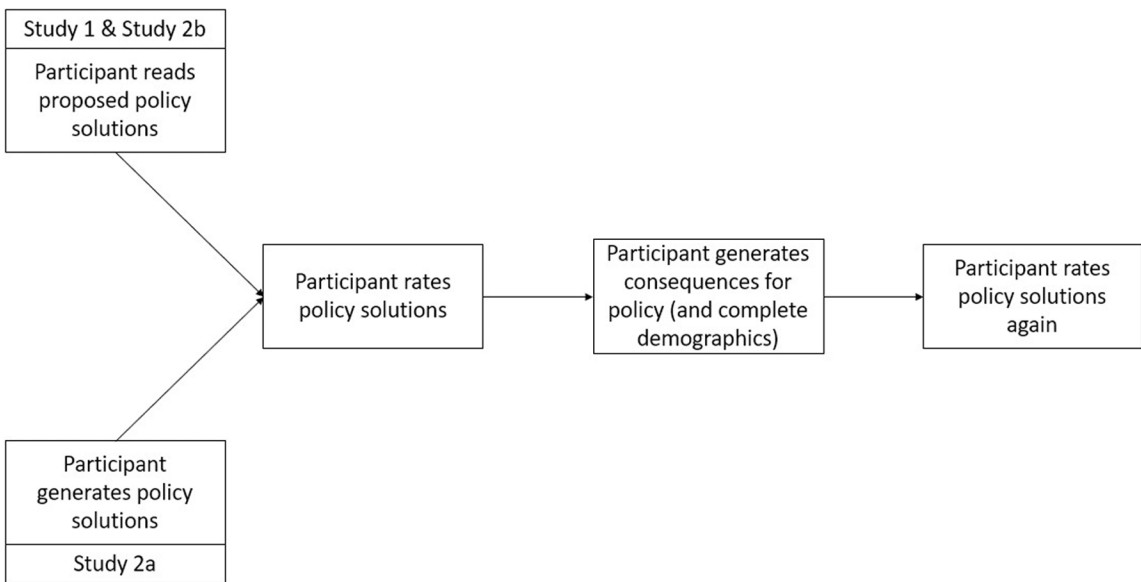

**Fig 1. Study Design for Studies 1, 2a, and 2b.** The above figure shows the general outline of the paradigm used throughout the studies in this paper. In general, participants began by rating policy solutions (the source of these policy solutions varied by study). Subsequently, participants were asked to generate two positive and two negative consequences for those same policy solutions. Finally, participants again rate policy solutions.

## Results

***Consequence neglect.*** Across all contexts, the average rating shift was 0.2 points downwards. A pairwise t-test reveals this shift to be significant ($t(664)=6.52$, $p < .01$, $d_{RM} = -0.244$). Results of rating shifts for the different policy contexts are shown in Table 1. Notably, every context tested was consistent with the pattern that consequence neglect would predict. Pairwise t-tests showed that this downward shift was statistically significant for three out of the six policy evaluation domains and also for the self-generated policy. Overall, these results show us evidence that a consequence generation task leads to a subsequent downgrade in ratings.

Of course, it was important to determine whether the order that consequences were generated (positive vs. negative consequences) had any bearing on the observation of this downward shift. If order effects were observed, then the results could be an artifact of priming (i.e., the most recently generated consideration having the most weight), rather

**Table 1. Mean Rating Shifts in Study 1.**

| Context | Average Rating Shift |
|---|---|
| Salesforce motivation | −0.07 |
| Neighborhood reinvigoration | −0.09 |
| Cosmetic surgery regulation | −0.12 |
| Corporate controversy | −0.24* |
| Minimum wage regulation | −0.26* |
| Teacher incentives | −0.28* |
| Office supply protection (Self-generation task) | −0.35* |

The above table displays the average rating shift for each context of Study 1. Rating shifts were calculated by subtracting the first rating from the second rating. As a result, negative rating shifts correspond to a rating revision downwards.

*p <.05

than evidence of consequence neglect. However, across each of the seven contexts, we found no significant difference between participants who generated positive consequences first and those who generated negative consequences first (all $p$s >.2). This provides evidence that the order in which consequences were generated did not influence the subsequent rating of a policy.

Despite the fact that people were fully capable of coming up with negative consequences, these downward shifts (four of which were significant) suggest that they may not have been considered in the first round of ratings. Examining some of the generated consequences sheds light on how easy some of these are to call to mind. For example, for the teacher incentive prompt, some commonly generated negative consequences were "Teachers teach to the test," "teachers change test scores," and "Teachers may feel stressed if students don't perform well" (A complete list of consequences as well as full datasets for all of our studies can be found in our Web Appendix which is in the files tab of the following OSF link: https://osf.io/8jrew/). It is not a matter of whether people are capable of generating consequences, but rather a matter of whether they attempt to do so.

***Individual differences.*** We began our scale analysis by appropriately reverse coding and summing across our scale items to make a composite score such that higher scores were associated with greater consequence neglectfulness. Our scale yielded fairly reliable measurements (Chronbach's $\alpha$ =.785). However, while the measure was reliable, it was not meaningfully associated with post consequence generation rating shifts as we had hypothesized. In fact, none of the individual differences that we measured were meaningfully related to observed levels of consequence neglect, including participants' CRT scores. Summary statistics describing these null results are shown in Table 2. This suggests that there is no "general consequence neglectfulness" trait that differs between individuals, (or to the extent that there is, such a trait not associated with demographics, measures of reflective thought, or measures of introspective awareness of one's tendency to ignore consequences).

## Discussion

Study 1 illustrated that a consequence generation task leads to a subsequent downward revision in one's evaluation of a given policy. We showed that the same trends occurred across all seven domains tested, and that the results were not an artifact of order effects. However, we did not find any evidence of associations between these downgrades and any individual difference measures, including measures of cognitive reflection, educational attainment, or self-reported tendency to think about consequences.

One intriguing observation was that people showed larger consequence neglect for self-generated solutions to problems than when evaluating solutions generated by others. However, there are (at least) two important caveats to

**Table 2. Null Relationships between Frequency of Downward Revisions and Individual Difference Measures in Study 1.**

| Demographic | Test Statistic |
| --- | --- |
| Age | $F(1,93) = 1.40$, $p =.24$ |
| Gender | $F(3,91) = 0.77$, $p =.51$ |
| Race | $F(4, 90) = 0.22$, $p =.93$ |
| Education level | $F(5,89) = 1.84$, $p =.11$ |
| Income level | $F(1,93) = 0.34$, $p =.56$ |
| Political Orientation | $F(1,93) = 1.10$, $p =.30$ |
| CRT Score | $F(1,93) = 1.25$, $p =.27$ |
| Novel Individual Difference Scale | $r(94) =.14$, $p =.17$ |

The above table reports the test statistics from ANOVAs (first seven rows) and a regression (last row) examining whether a given individual difference measure was significantly associated with the number of downward revisions an individual makes. There were no significant relationships. For the case of the novel individual difference scale, divided individuals into two groups via a median split.

this finding. First, there was only one self-generated solution domain (office supply theft), and it is possible that the specific domain tested happened to evoke stronger consequence neglect than the other domains tested (i.e., stimulus sampling issues [19]). Second, the other-generated solutions were all generated by the research team. Given that the research team was not blind to the hypotheses when generating stimuli, it is possible that the experimenter-generated stimuli could artificially inflate rates of negative consequence neglect. Although the data patterns in Study 1 suggest that if anything, the opposite is true, that could be an artifact of flawed stimulus sampling. While we explored seven different policy contexts, for the six vignettes with other-generated solutions only a single policy solution was proposed/evaluated. The results would be more compelling if there were a wider range of policy solutions proposed, and those proposed policy solutions were generated by individuals who were blind to the goals of the study.

Another potential issue with Study 1 is the dependent measure. Participants were asked to evaluate the likely *effectiveness* of a solution. This question may not have fully captured the concept we were hoping to measure: a solution can be extremely effective at resolving the problem at hand while still having problematic externalities. Participants might have neglected consequences because they thought that consequences weren't germane to the specific question being asked. As such, in Study 2a we asked participants to evaluate the overall *favorability* of the solution, which is less likely to be interpreted as solely about how well the solution deals with the proximal problem. Study 2a was designed to resolve these issues.

## Study 2a: Consequence neglect for self-generated solutions

Study 2a differs from Study 1 in several ways. First, participants in Study 2a generated their own solutions rather than being provided with experimenter-generated solutions. It may be that in the process of generating solutions, people think more deeply, which might lead consequences to come to mind. Alternatively, the goal-oriented nature of developing a policy to solve a problem may actually exacerbate consequence neglect by motivating people to focus on the problem at hand rather than the unintended results of their solutions.

Secondly, the participants were asked to evaluate the favorability of a policy, rather than the effectiveness of a policy. Third, we used several new scenarios to further generalize the results to a more diverse set of policy domains. Finally, as our measure of individual differences in consequence neglect had proven to be uninformative in Study 1, it was removed from Study 2a to shorten the experiment and save time.

### Method

*Participants.* Subjects were recruited from Amazon Mechanical Turk for a flat rate payment of $3.00 based on the estimated time to complete the survey. We intended to recruit 200 participants. As with Study 1, we took two steps to ensure data quality: 1. Upfront attention checks were used to screen out potential bots or inattentive subjects. 2. Before exiting the survey, participants were given an open-ended free response question that sought to differentiate humans from bots. After accounting for these screens, our final sample size for analysis consisted of 181 participants (48% female, mean age = 39 years).

*Procedure.* Similar to Study 1, after participants completed these procedures, they were faced with five hypothetical problems. The domains of these problems were: motivating a high schooler who has stopped doing homework, raising money for new equipment at a gym, responding as a CEO to company controversy, preventing office supply theft, and alcohol regulation on a college campus. However, unlike most of the problems in Study 1, participants were not provided with a solution to evaluate.

Instead, problems were presented one at a time in random order, and participants generated a solution for each of them. After completing this solution generation task, each participant was shown the solutions that s/he had proposed and asked to evaluate how favorable each proposed policy was on a 5-point Likert scale (1=Not at all favorable, 5=Extremely

favorable). From this point onwards, this experiment parallels the procedure of Study 1. Participants completed demographic information (which no longer included our developed scale or the CRT, given the lack of association between these scales and consequence neglect rates in Study 1). Then they completed a consequence generation task and second round evaluation exactly as in Study 1.

## Results

Across all contexts, there was an average downward revision of 0.19. A pairwise t-test reveals this shift to be significant ($t(904) = 6.82$, $p <.01$, $d_{RM} = −0.225$). Results across all domains are shown in Table 3. Notably, pairwise t-tests showed significant shifts in four out of the five policy evaluation domains. The fifth domain showed a non-significant shift in the opposite direction. Despite developing these solutions themselves, participants exhibited the same general downshift in ratings in this study as Study 1.

As in Study 1, we tested for order effects. We showed no statistically significant evidence for order effects, although the gym context showed a marginally significant order effect in which participants gave a slightly higher final rating when they listed negatives first and positives last, the opposite pattern than we would expect if the results were driven by priming ($t(175) = -1.87$, $p =.063$, $d = 0.28$; all other $p$s >.2). Given the lack of significance across contexts, we conclude that the order in which consequences were generated did not influence the subsequent rating of a policy.

## Discussion

Similar to Study 1, Study 2a illustrated that a consequence generation task leads to a subsequent downward revision in the evaluation of a given solution. These results extend the findings of Study 1 by demonstrating consequence neglect across several new scenarios. More importantly, Study 2a shows that the phenomenon persists even for self-generated policies.

Despite the observation that consequence neglect seems to occur both when evaluating self-generated and other-generated policies, one question that lingers is the relative magnitude of the effects for generators versus pure evaluators. The results from Study 1 and Study 2a are not directly comparable because different stimuli were used between the studies. Although in Study 1 the effect was larger for the self-generated solution, it is hard to draw strong inferences from this result because 1) only one self-generated solution was tested (and perhaps that scenario was particularly susceptible to consequence neglect), and 2) the other-generated solutions were created by experimenters who were not blind to experimental hypotheses and thus could have inadvertently introduced bias or demand into the stimuli.

Fortunately, Study 2a yielded a rich set of diverse proposed policy solutions. These stimuli can be used to speak to cross-stimulus generalizability [19]. Thus, to answer the question of the relative prevalence of consequence neglect for

**Table 3. Mean Rating Shifts in Study 2a.**

| Context | Average Rating Shift |
|---|---|
| High schooler motivation | −0.22* |
| Gym renovation | −0.17* |
| Corporate controversy | −0.39* |
| Office supply protection | −0.28* |
| Alcohol regulation | 0.08 |

The above table displays the average rating shift for each context of Study 2a. Rating shifts were calculated by subtracting the first rating from the second rating. As a result, negative rating shifts correspond to a rating revision downwards.

*$p <.05$

self- vs. other-generated policies, in Study 2b we used the policies generated by the participants in Study 2a as stimuli. This allows for an apples-to-apples comparison between rating shifts for policy-generators and policy-evaluators.

## Study 2b: Consequence neglect for other-generated solutions

Study 2b was designed for two purposes: (1) to test whether consequence neglect tends to be larger for generators or evaluators and (2) to replicate the findings of Study 1 with a more diverse and unbiased set of stimuli. Specifically, given that our initial policies were developed with obvious negative externalities in mind, using this form of stimulus sampling ensures that a consequence generation task can shift people's policy evaluations without unconsciously putting our thumbs on the scale. Study 2b achieved these goals by utilizing solutions generated by participants in Study 2a as stimuli for participants in Study 2b.

### Method

*Stimuli selection.* Participants were presented with the same five problems as in Study 2a. However, rather than generating their own solutions, they evaluated solutions that were proposed by the participants in Study 2a. In some cases, this required editing the wording to make the solution flow (without changing the meaning). For example, many solutions in Study 2a were generated in first person (e.g., "I would make an inventory sheet to sign out supplies when needed"). We edited these to remove reference to the self (e.g., "make an inventory sheet to sign out supplies when needed"). Three RAs independently evaluated whether each prompt should be selected for inclusion in Study 2b. Unclear, confusing, irrelevant, and ambiguous solutions were removed (e.g., "400"). In addition, multi-barreled solutions (e.g., "Bribe him, or take away things he enjoys until he does it.") were removed because it would be unclear whether participant evaluations reflect the first barrel, the second barrel, or both. Table 4 displays the number of proposed solutions from Study 2a that were included in Study 2b for each of the five domains.

*Participants.* Subjects were recruited from Amazon Mechanical Turk for a flat rate payment of $2.25 based on the estimated time to complete the survey. We intended to recruit 124 participants. As with the prior two studies, we took two steps to ensure data quality: 1. Upfront attention checks were used to screen out potential bots or inattentive subjects. 2. Before exiting the survey, participants were given an open-ended free response question that sought to differentiate human from bot. After accounting for these screens, our final sample size for analysis consisted of 113 participants (35% female, mean age = 36 years).

*Procedure.* After participants completed these procedures, they were faced with the same five hypothetical problems as in Study 2a. However, rather than generating their own solutions to these problems, participants read and evaluated solutions written by participants from Study 2a. Each participant only saw one solution per context; however, across all participants approximately 500 different solutions were evaluated. Aside from the use of participant-generated stimuli, this survey was identical in procedure to Study 2a.

**Table 4. Number of solutions selected for each context.**

| Context | Number of Solutions |
| --- | --- |
| High schooler motivation | 101 |
| Gym renovation | 119 |
| Corporate controversy | 99 |
| Office supply protection | 124 |
| Alcohol regulation | 84 |

The above table displays the number of participant-generated solutions from Study 2a that were successfully adapted for use in Study 2b.

## Results

***Consequence neglect.*** Similar to other studies, we found a significant mean revision downwards of 0.28 ($t$(564) = 8.56, $p$ <.01, $d_{RM}$ = -0.353). Our results of rating shifts for each domain are shown in Table 5. In this study, pairwise t-tests showed significant shifts in each of our five policy evaluation domains.

As with the prior studies, for four of these contexts, we found no significant order effects ($ps$ >.6). The fifth context (CEO corporate controversy) did show a significant order effect ($t$(111) = 2.03, $p$ =.045, $d$ = 0.38) in which participants gave a significantly higher final rating when they listed negatives first and positives last (the opposite trend from what we would expect if there was a priming confound). However, because this order effect was not significant for this scenario in prior studies, and no other scenarios showed order effects, we believe this result is likely a false positive.

***Comparison to Study 2a.*** By using stimuli that were generated by participants during Study 2a, we are able to make an apples-to-apples comparison between self-generated and other-generated solutions with regard to the prevalence of consequence neglect. However, since each domain had a different number of stimuli, this means some stimuli were displayed to more than one participant. For instance, given that there were 124 stimuli in the office supply context but only 84 in the alcohol regulation context, there were at least 40 duplicates in the alcohol regulation context. In reality, the number of duplicates in each context is higher than the theoretical amount, due to randomization of stimulus presentation. Additionally, as some participants were screened out of the survey for failing a bot check, whatever stimuli that they would have seen would not have been included in the final sample, so it is possible that there were some stimuli not seen at all. Duplicates were identified and removed in a random fashion to avoid the creation of any potential bias (i.e., for each stimulus that had been seen by two participants, one of the two was randomly removed from the sample before final analysis). This allowed for a one-to-one comparison for each unique solution, one from a participant in Study 2a (self-generated), and the other from a participant in Study 2b (other-generated). We ran a re-analysis of Study 2a, considering only the subset of responses that were used in Study 2b. The results aligned with the original analysis: there was a significant mean revision downwards of.20 ($t$(433) = 4.93, $p$ <.01, $d_{RM}$ = −0.24). Similarly, the re-analysis of Study 2b evaluations with duplicates removed showed a significant mean revision downwards of.28 ($t$(433) = 7.04, $p$ <.01, $d_{RM}$ = −0.34). Thus, the findings are robust when considering only the subset of policies included in both studies.

Because participants saw effectively identical prompts, we could use a pairwise t-test to explore whether rating shifts were larger for self-generated or other-generated solutions. On aggregate, we found that there was no significant difference in ratings shifts between the generators from Study 2a and the evaluators from Study 2b ($t$(433) = 1.28, $p$ =.20, $d$ = 0.12). There was no consistent trend across scenarios; for two scenarios the rating shifts were higher in Study 2a; for two scenarios the rating shifts were higher in Study 2b, and for the final scenario the rating shifts were the same. This suggests that the consequence generation task had similar effects on both self- and other-generated solutions. However, unsurprisingly, people did rate their own solutions higher than others' solutions. All ratings for the solutions in this comparison set are shown in Table 6.

**Table 5. Mean Ratings Shifts by Context in Study 2b.**

| Context | Average Rating Shift |
| --- | --- |
| High schooler motivation | −0.33* |
| Gym renovation | −0.26* |
| Corporate controversy | −0.15* |
| Office supply protection | −0.48* |
| Alcohol regulation | −0.19* |

The above table displays the average rating shift for each context of Study 2b. Rating shifts were calculated by subtracting the first rating from the second rating. As a result, negative rating shifts correspond to a rating revision downwards.

*$p$ <.05

**Table 6. Ratings by Context in Study 2a and Study 2b.**

| Context | Study 2a Rating 1 | Study 2a Rating 2 | Study 2a Rating Difference | Study 2b Rating 1 | Study 2b Rating 2 | Study 2b Rating Difference |
|---|---|---|---|---|---|---|
| High schooler motivation | 3.85 | 3.56 | −0.29* | 3.14 | 2.85 | −0.29* |
| Gym renovation | 3.68 | 3.44 | −0.24* | 3.08 | 2.89 | −0.19* |
| Corporate controversy | 3.73 | 3.33 | −0.40* | 2.8 | 2.64 | −0.16* |
| Office supply protection | 3.57 | 3.37 | −0.20* | 3.04 | 2.57 | −0.47* |
| Alcohol regulation | 3.22 | 3.36 | 0.14 | 3.07 | 2.89 | −0.18* |

The above table displays all ratings for Study 2a and Study 2b. Notably, ratings were higher for participants in Study 2a, but despite this higher starting point, the differences between ratings across studies were largely comparable.

*significant from 0 at p <.05

## Discussion

Consistent with the prior two studies, Study 2b illustrated that a consequence generation task leads to a subsequent downward revision in the evaluation of a solution. Given the varied stimuli in this study – literally hundreds of participant-generated solutions - we are more confident in the cross-stimulus generalizability of the results.

Considered together, Study 2a and Study 2b allow us to make an apples-to-apples comparison between rating shifts for self-generated vs. other-generated policy solutions, and we find that consequence neglect is robust across this dimension. This suggests that consequences are equally neglected during the generation process and the evaluation process, unless attention is explicitly called to externalities. This aligns with the focalism account of consequence neglect; consequence neglect could happen during any part of the policy-making process for which consequences are not focal to solving the problem at hand.

While this study provides yet another piece of evidence consistent with the presence of consequence neglect, there are two alternative explanations to the pattern of results that are worth exploring. Rating shifts may not be due to the consequence generation itself. Instead, the shift could be attributable to 1) merely re-rating the policy or 2) merely thinking more about the solution (content unrelated to consequences). The first account suggests that doing the same task twice leads people to become more negative, and thus evaluations will tend to go down irrespective of any task happening in between the two evaluations. The second account suggests that it is not thinking about consequences per se that leads to downward shifts in evaluations, but that more time, fatigue, or depth of processing generally leads people to identify flaws in a policy and lower evaluations. These possibilities are addressed in Study 3.

## Study 3: Ruling out confounds

### Method

*Participants.* Subjects were recruited from Amazon Mechanical Turk for a flat rate payment of $2.25 based on the estimated time to complete the survey. We intended to recruit 300 participants. As with the prior two studies, we took two steps to ensure data quality: 1) Upfront attention checks were used to screen out potential bots or inattentive subjects. 2) Before exiting the survey, participants were given an open-ended free response question that sought to differentiate human from bot. After accounting for these screens, our final sample size for analysis consisted of 270 participants (45% female, mean age = 41 years).

*Procedure.* Study 3 shares a similar structure to prior studies, using the same materials as those used in Study 2b. Participants were randomly assigned to one of three conditions: consequence generation (a replication), thought generation, and re-evaluation only. Participants in each condition began the survey in an identical manner by evaluating each of the five proposed policies. The consequence generation condition was identical to Study 2b, and serves as a replication and comparison standard for the two new conditions.

In the thought generation condition participants followed a similar procedure to the consequence generation condition. However, instead of being asked to generate consequences for each policy, they were asked to, "Please take a moment to reflect on how this plan could be implemented and suggest initial steps you would take to put this plan into motion." We chose implementation to have participants reflect upon as it is not inherently positively or negatively valenced. If merely thinking further about the policy (without making consequences focal) yields similar patterns of results to earlier studies, then the findings may have little to do with consequences, but are rather due to negative shifts after deliberation more generally.

In the re-evaluate condition, after participants submitted their first round of ratings, they completed the demographics survey, and then immediately submitted their second round of ratings. If mere re-evaluation leads to downward shifts in ratings, then this condition should yield the same effects as previous studies, despite not having compelled participants to attend to consequences. Notably, these conditions also allow us to rule out fatigue as a possible alternative explanation. If fatigue were driving negative downshifts in policy evaluations, then we would expect to see significant downshifts in the consequence and thought generation conditions (which are identical in length) compared to the control condition (which is shorter since participants do not spend time brainstorming and writing information about the policy).

Participants in all three conditions ended the survey in an identical manner by re-evaluating each of the five proposed policies. A summary of this experiment's design is shown in Fig 2.

## Results

Similar to prior studies, our consequence generation condition yielded a downshift in ratings. The downshift in the consequence generation condition was −0.21 which is significantly different from the downshift of −0.01 in the thought generation condition ($t(868) = 3.664$, p <.01, $d = 0.25$). The consequence generation condition's downshift of −0.21 was also significantly different from the downshift of −0.04 in the re-evaluation only condition ($t(766) = 3.662$, $p <.01$, $d = 0.25$). The downshift in the thought generation condition and the downshift in the re-evaluation only condition were not significantly

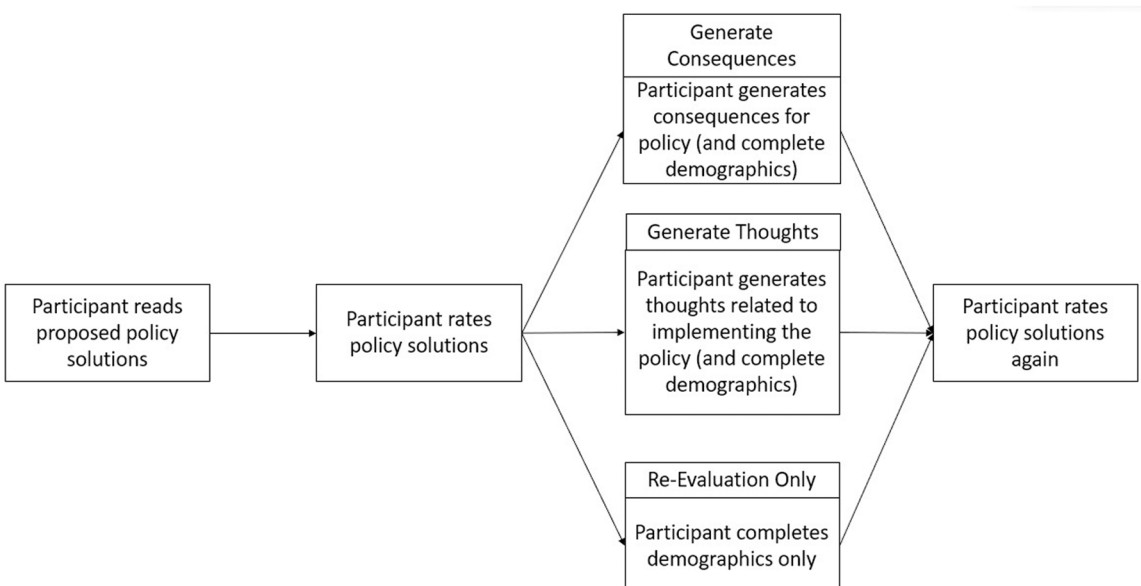

**Fig 2. Study 3 Format.** The above figure shows the outline of the paradigm used in Study 3. Upon entry into the survey, all participants were asked to read and rate solutions in the five different policy domains from Study 2b. After submitting this initial rating, participants were randomly assigned to one of three conditions. After participating in the task relevant to their condition, participants submitted a final rating for all policy solutions.

different from one another (*t*(755) = 0.63, *p* =.26, *d* = 0.05; for a complete breakdown of rating shifts by condition and context, see Table 7). Taken together, these results suggest that the process of generating consequences leads to an increased downshift relative to thinking about how to implement the policy or merely re-evaluating the policy.

## General discussion

Across four studies we find evidence that in general, a consequence generation task leads people to evaluate policies as significantly worse than they did prior to the task, even though participants generated both positive and negative consequences. This downward shift suggests that negative consequences are typically not considered unless attention is explicitly drawn to them. This downshift was present for both self-generated and other-generated policies, for approximately 500 different policy solutions, across 10 different policy contexts, for two different dependent measures, and this downshift was not present when participants were merely re-evaluating policies or thinking about other aspects of the policy, such as implementation plans. This suggests that individuals often fail to consider consequences even when the policies are both 1) relevant to the policy evaluation at hand and 2) knowable to the individuals doing the evaluation.

It is also worth noting that in these studies participants were required to list both negative *and positive* consequences of various policies. Thus, the findings cannot be due to experimenter demand effects (i.e., participants guessing the hypothesis and being more negative in an attempt to respond to social cues), or to differential priming of negative vs. positive features biasing evaluations. Indeed, we found no consistent evidence of order effects – regardless of whether people listed positive or negative consequences first, they downgraded their evaluation of a policy after thinking about the consequences.

Given the importance of accurately assessing solutions to problems, there is value to be found in interventions designed to mitigate consequence neglect. Research on "consider the opposite" strategies suggests that explicitly prompting individuals to think about alternative scenarios can reduce bias [20]. Analogously "consider the consequences" strategies could attenuate the consequence neglect bias. Broadly speaking, application of behavioral public policy approaches and nudge-based interventions could help decision-makers become more aware of neglected consequences [14,21].

## Limitations and future directions

These studies were not incentive compatible. Given that these were hypothetical issues with no real stakes, participants may not have been fully engaged with initial evaluations. In the real world, when people are considering important policies with material impact rather than hypothetical consequences, people may be more motivated and think more deeply when evaluating policies. Under such conditions, consequence neglect may be diminished. Thus, while the present studies were motivated by real-world examples consistent with consequence neglect, it will be important for future studies to investigate the phenomenon in the wild and under incentive compatible conditions.

**Table 7. Mean Rating Shifts by Context in Study 3.**

| Context | Consequence Generation | Thought Generation | Re-Evaluation Only |
|---|---|---|---|
| High schooler motivation | −0.39* | 0.05 | 0.02 |
| Gym renovation | −0.27* | 0.00 | -0.04 |
| Corporate controversy | −0.22* | 0.01 | 0.03 |
| Office supply protection | −0.21* | −0.03 | −0.18 |
| Alcohol regulation | 0.01 | −0.05 | −0.03 |
| Average Across All Contexts | −0.21* | −0.01 | −0.04 |

The above table displays mean rating shifts by condition for each context across all contexts. Notably, the consequence generation condition was associated with more significant downshifts in ratings than either of the other two conditions.

*significant from 0 at p <.05

These studies were conducted on a WEIRD population [22]. It may be that cultural differences in how policies are evaluated or what is considered focal could impact the extent to which consequence neglect is observed across different populations. In Study 1, we found no evidence that individual differences or demographic differences (including education, race, gender, and political orientation) moderate the effect of consequence neglect. However, there could be other moderators for future research to examine. For example, a policy's personal relevance, a decision maker's domain expertise, or a decision maker's profession may affect results. In our studies, we did not have any a-priori exclusion criteria, such as holding political office or pursuing a degree in political science or law. Showing that consequence neglect persists in such populations would be a useful extension of this research to show that the findings could have real world policy impact.

It is possible that in the present studies both positive and negative consequences get overlooked, and the pattern of results arises because the negative consequences exert a bigger influence on the re-evaluation than the positive consequences do. However, this possibility is not mutually exclusive with our consequence neglect account. Specifically, we articulate that consequences that are non-proximal to the goal will not be focal, and thus, will be underweighted in the policy evaluation process. In general, positive consequences are more likely to be proximal to the goal of the policy, (after all, solving a problem was the reason that the policy was created in the first place) and as a result, they are more likely to be considered during the evaluation of a policy. However, one could certainly conceive of situations in which there exist positive consequences that are non-focal, and thus will go underweighted without explicit generation.

Much research on effort reduction strategies in the heuristics and biases literature has discussed the effects of cognitive load and time pressure on effortful processing (for a review, see [23]). In the present studies, participants were not under load, nor under particularly stringent time constraints; we would expect that such pressures would only exacerbate consequence neglect, as it would reduce participant cognitive capacity to consider non-focal information in problem solving. Of course, in the real-world policy makers, business managers, and everyday people faced with problems are operating under conditions of stress, high load, and with limited time. Thus, to understand the magnitude of consequence neglect in more naturalistic settings, it would be worthwhile to explore the construct under load and time pressure.

Of course, not every single person will be consequence neglectful in every single problem-solving context. In some contexts, for some people, consequences may be more or less proximal to the goal at hand. In circumstances where consequences are focal to the goal, they would be more likely to be spontaneously considered during policy evaluation. However, in circumstances where consequences are less naturally focal to the goal, these consequences may be neglected. Identifying what makes an element of a solution "focal", and why, is an important future question for research on consequence neglect, and focalism more generally. Until those questions are answered, we are forced to rely on our intuitions of what counts as focal, which is imprecise and makes it difficult to predict to which contexts these results will generalize.

While we have focused on focalism as a mechanistic explanation for consequence neglect, we acknowledge that there are many reasons across many levels of analysis that would lead to consequences being neglected in real world policy domains. For example, organizational structure or political/ideological constraints could lead to policies that ignore negative consequences [3]. Our current work serves as an existence proof that consequences are neglected even in the absence of such systemic and structural barriers to attending to consequences. However, those factors are important, and future work should explore how individual level mechanisms may interact with systems level mechanisms in preventing optimal problem solving.

***Fluency.*** Processing fluency refers to the metacognitive ease people experience when processing information. This metacognitive ease has been shown to influence a wide variety of domains [24]. In a seminal work, Schwarz et al. [25] asked participants to recall examples of themselves acting assertively. Under one condition, participants were asked to generate six such examples (an easier task), and in the other, participants were asked to generate twelve examples (a more difficult task). Subsequently, they rated how assertive they believed themselves to be. One might think that people would rate themselves higher in response to generating more examples. However, participants rated their assertiveness based on the difficulty they had generating those examples. Ultimately, it was the ease of example generation that drove the ratings,

with the participants who generated six examples generally rating themselves as more assertive than those participants who generated twelve examples.

Such a fluency effect may be able to account for part of the experimental results that we observed. Such an account would hinge upon the subjective ease with which people can generate positive and negative consequences. In each of our studies, we asked participants to generate two positive and two negative consequences. In general, the first positive consequence was obvious: solving the problem at hand. However, the second positive consequence is likely more difficult to generate. Analogous to Schwarz et al. [25], by requiring more positive consequences to be generated, the task becomes more metacognitively taxing, and as a result, the policy is given lower evaluations. Meanwhile, generating two negative consequences may be relatively easier, as most policies have some associated costs. While this is a possible alternative explanation for our results, it seems unlikely to be the driving mechanism. Considering that earlier studies demonstrating fluency effects asked for six examples in the *fluent* condition (and twelve in the disfluent condition; e.g., Schwarz et al. [25]) the idea that asking for a single non-focal positive consequence would be enough to yield the negative evaluations observed in the present manuscript seems somewhat implausible. Nevertheless, future experiments could attempt to rule out this explanation by manipulating the number and type of consequences that participants are asked to generate.

### Implications for policy

While it is an empirical question as to how much consequence generation influences policy outcomes, we argue that it will likely, on average, improve them. This is because of the implications of consequence neglect on cost-benefit analysis, which is widely considered the gold standard for policy evaluation [26]. Sunstein argues for cost-benefit analysis not only from an economic perspective but from a cognitive perspective as well, noting that it helps overcome a variety of predictable problems in individual and social cognition, including overreliance on heuristics and biases. While there is little doubt that people consider the benefits of a policy, if people do not consider the unintended consequences, then they will tend to underestimate the costs. This would undermine the value of cost-benefit analysis. In many of our studies, participants named consequences that were clearly crucial to successful evaluation of a policy. Indeed, as the cobra effect exemplifies, there exist many historical policies for which predictable consequences led to the failure of well-intentioned interventions.

Of course, there may be cases when calling explicit attention to consequences results in costs being overweighted to the point of delaying or outright cancelling deployment of an otherwise desirable policy. If the focal problem was particularly urgent or dangerous this could be undesirable; often the benefits of a policy legitimately outweigh the costs of negative externalities. The irony of the need to draw attention to the possible negative consequences of interventions to overcome consequence neglect is not lost on us. However, especially for issues that are not particularly time sensitive or have multiple, highly variant solutions, explicit consideration of negative consequences could overcome consequence neglect and lead to better long-term outcomes.

### Conclusion

Our studies provide robust evidence that individuals systematically underweight or ignore non-focal consequences when evaluating solutions. The consistency of this effect across multiple contexts, including self-generated and pre-designed solutions, suggests that consequence neglect is a general cognitive bias with implications for decision-making and public policy. It may be discouraging to consider that consequences may often go neglected. However, our results also speak to the idea that explicitly articulating the consequences makes them focal to the policy evaluation. Thus, consequence neglect is a pervasive but correctable bias.

The implications of this work extend beyond the laboratory. In policymaking, business strategy, and everyday decision-making, structured approaches that encourage consequence consideration—such as cost-benefit analysis, pre-mortem exercises, and scenario planning—could significantly improve the quality of decisions. Our results suggest that

decision-makers do not fail to recognize consequences because they are incapable of doing so, but rather because these consequences are not naturally focal in the policy evaluation process. By embedding structured consequence evaluation into institutional processes, organizations and policymakers can create more resilient, forward-thinking solutions that anticipate and mitigate unintended side effects before they arise.

Future research should explore what types of problems are most likely to be associated with consequence neglect as well as what sort of individual differences may predict consequence neglect. This could further inform scalable, real-world interventions to help individuals and policymakers proactively account for consequences in their decision-making. Whether through policy design frameworks, behavioral nudges, or training programs, there is an opportunity to improve decision-making at both individual and societal levels. By recognizing and addressing consequence neglect, we can move toward a world where policies are not only well-intentioned but also well-executed, creating better, more sustainable outcomes for all.

The stakes are high: for policymakers tasked with solving today's challenges, the consequences of ignoring consequences are like swallowing the proverbial horse, with well-meaning policies that are dead, of course.

## Author contributions

**Conceptualization:** Christopher Rodriguez, Daniel M. Oppenheimer.

**Data curation:** Christopher Rodriguez.

**Formal analysis:** Christopher Rodriguez.

**Funding acquisition:** Daniel M. Oppenheimer.

**Investigation:** Christopher Rodriguez.

**Methodology:** Christopher Rodriguez, Daniel M. Oppenheimer.

**Project administration:** Christopher Rodriguez.

**Resources:** Christopher Rodriguez.

**Software:** Christopher Rodriguez.

**Supervision:** Christopher Rodriguez, Daniel M. Oppenheimer.

**Validation:** Christopher Rodriguez.

**Visualization:** Christopher Rodriguez.

**Writing – original draft:** Christopher Rodriguez.

**Writing – review & editing:** Christopher Rodriguez, Daniel M. Oppenheimer.

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
