## [Decision Letter · Decision Letter 0]

31 Jan 2025

PONE-D-24-48769Side Effects May Include: Consequence Neglect in Generating SolutionsPLOS ONE

Dear Dr. Rodriguez,

Thank you for submitting your manuscript to PLOS ONE. After careful consideration, we feel that it has merit but does not fully meet PLOS ONE’s publication criteria as it currently stands. Therefore, we invite you to submit a revised version of the manuscript that addresses the points raised during the review process.

We look forward to receiving your revised manuscript.

Kind regards,

Tobias Otterbring

Academic Editor

PLOS ONE

Additional Editor Comments:

Dear authors,

Your manuscript was sent to three reviewers with considerable knowledge and expertise in your topic domain. I have now received the review reports from two of them. As these reviewers are fairly consistent in their feedback, I decided to make a decision without having you wait for the final one. One reviewer recommends a major revision, while the other one recommends a minor revision. Both the reviewers raise many valid concerns related to your conceptualization, construct definitions, treatment of confounding factors, transparency aspects, and analytic decisions. Based on my own reading of the manuscript, I am willing to move your manuscript into another round of reviews. Given the magnitude of some of the raised issues, this will be a major revision, albeit a revision with a relatively clear path toward publication as long as you meticulously address all the valuable feedback points from the reviewers in your revision. One reviewer requests additional data in at least one new study. While I am convinced that such a study will bolster the quality and contribution of your empirical package, I am agnostic as to whether you *must* include more data as part of this submission. Thus, if you believe that you can compellingly address some of the issues identified through your existing data, you might be able to convince this reviewer that your work is still worthy of publication even without added empirical evidence. That said, the decision whether or not to collect additional data is ultimately your call.

Kind regards,

Tobias Otterbring

Handling Editor, PLOS One

Reviewers' comments:

Reviewer's Responses to Questions

**Comments to the Author**

1. Is the manuscript technically sound, and do the data support the conclusions?

Reviewer #1: Yes

Reviewer #2: Partly

2. Has the statistical analysis been performed appropriately and rigorously? 

Reviewer #1: Yes

Reviewer #2: No

3. Have the authors made all data underlying the findings in their manuscript fully available?

Reviewer #1: Yes

Reviewer #2: No

4. Is the manuscript presented in an intelligible fashion and written in standard English?

Reviewer #1: Yes

Reviewer #2: Yes

5. Review Comments to the Author

Reviewer #1: In this manuscript, the authors investigate the concept of consequence neglect, understood as the tendency of policymakers to overlook (negative) consequences of policies when generating and evaluating said policies. The results of four online experiments (Ntotal > 600) show convincing and robust evidence for the proposed effect, demonstrating that individuals’ perceptions (in terms of effectiveness, Study 1, or favorability, Studies 2a-3) of potential solutions deteriorate when being instructed (vs. not) to generate and consider consequences of these solutions. While the concept may not be entirely new (considering the closely related/similar concepts of focalism and opportunity cost neglect mentioned by the authors), the current research nicely extends previous findings to the context of policies in various domains (e.g., education, organizational, urban planning, public health).

The manuscript is well-written, has a clear structure, and displays transparency. It is based on studies that build on each other in a meaningful way and address several limitations (specifically, Study 3), further demonstrating the robustness of the findings.

Overall, there are only a few minor aspects the authors could address for clarification and to further improve their manuscript. These issues will be briefly outlined below (in no particular order).

#1. Literature/References

The authors could consider introducing additional relevant references, as the current literature base is relatively scarce (i.e., 13 references overall, including introduction/theory and discussion). This does not mean that the authors should add sources just to increase the number of references, but they may consider slightly extending their theoretical reasoning (e.g., from the literature on heuristics and biases) and/or discussion in the form of a few additional paragraphs and/or relevant sources.

#2. Effect Sizes

In addition to t- and p-values, the author should consider reporting effect sizes for their studies, as these are currently not included in the manuscript.

#3. Individual Differences

In Study 1, the authors investigated individual differences by using a novel, self-developed scale, the Cognitive Reflection Task of Frederic, and by looking at potential effects of socio-demographic characteristics. On page 12, they state that “none of the individual differences that we measured were meaningfully related to observed levels of consequence neglect (…) Summary statistics describing these null results are shown in Table 2”. While the column labelled “Test Statistic” of Table 2 (p. 13) may suggest that ANOVAs were conducted, the actual type of analysis it is not explicitly stated. This can be easily resolved.

In addition, I was wondering about the analysis regarding the relationship between the number of downward revisions and the Novel Individual Difference Scale – did the authors divide individuals in two groups based on their scores (e.g., low vs. high)? If so, what was the criterium (e.g., median split)? A correlation analysis might also have been appropriate considering that both variables were continuous.

#4. Excluded Stimuli

Overall, the authors are very transparent in their reporting. To further increase this transparency, they might consider including one or two examples of solutions that were excluded in the process of stimulus selection for Study 2b (i.e., examples for, “Unclear, confusing, irrelevant and ambiguous solutions were removed. In addition, multi-barreled solutions were removed”, p. 18f.).

#5. Discussion & Future Directions

In the discussion, the authors mention that “While we found no evidence that demographic differences (including education) moderate the effect of consequence neglect…” (p. 26). I was wondering whether this statement refers to the null effects reported for Study 1, or whether moderating effects of demographics were also tested for the other studies (but not explicitly mentioned). The authors could clarify this by briefly mentioning it in the results sections (or in the discussion).

In addition to socio-demographics, are there any other potential moderators (and maybe even mediators) the authors can think of that could be addressed in future studies? For example, personal relevance of the subject/domain of the policy under evaluation (or domain expertise), private/professional political engagement (in addition to political orientation, which was investigated in Study 1) may also affect the results. In direct relation to this: Did the authors apply any a-priori exclusion criteria, such as holding political office or pursuing a degree in Political Science or Law? If not, this could also be considered a limitation and could be pursued in a future study (e.g., comparing policymakers to “regular citizens”).

Furthermore, future studies could also investigate potential effects of external factors such as time pressure or cognitive load, as such factors are often discussed in the context of heuristics and biases.

#6. Conclusion

The current conclusion (p. 30) seems to be framed a bit negatively. I think the authors could consider explicitly mentioning the robustness of their findings in order to highlight a very positive aspect of their research.

#7. Missing Information in the Appendix / Missing List of Consequences?

On page 12, the authors state that “(A complete list of consequences can be found at OSF at: https://osf.io/8jrew/?view_only=13708524b2ab46889b0eea7bf790a24)“. Unfortunately, this link cannot be accessed directly, as one is directed to a page where one has to log in to OSF and request permission to access the file. In case there is a typo in the link and it is the same as the one leading to the overall Appendix file: I also checked the current Appendix file I was able to access on OSF (the first link provided in the manuscript), where I was able to find the instructions and self-generated solutions etc., but I did not find the list of consequences in this file.

#8. Inconsistencies in Formatting

There are some inconsistencies in formatting/style in the body of the text (e.g., inconsistent use of leading zeros) and in the reference list (e.g., use of italics for journal names, inconsistencies in capitalization of article titles, abbreviation of first names).

I am uncertain about citation guidelines of the journal, but if APA style is recommended, the authors may also check some deviations from APA in the body of the text (e.g., “Alpha” instead of “α“ on p. 12; test statistics are not in italics, e.g., ps and ts).

(Personal comment: I appreciate the insider reference to The Office in the Dunder Mifflin-Scenario :)

Reviewer #2: I really enjoyed reading your paper and I think you test an interesting and important phenomenon. However, there are several major issues that need to be addressed to improve the quality and clarity of your work. Below are my main concerns.

Major issues:

• The paper introduces the term "consequence neglect" but does not explain what it is and how it differs from existing concepts like "focalism in decision making" or "opportunity cost neglect" or “consideration of future consequences”. If this is really a new concept, you need to explain it clearly and discuss its relationship to prior research.

• You reference many constructs, such as focalism, opportunity cost neglect, and the illusion of explanatory depth, but the connections between them and your findings are unclear. This creates confusion instead of grounding the study in a solid theoretical framework.

• The Introduction includes fantastic examples (I really loved them!), but they do not align well with what you actually test. The song and the cobra example refer to a different phenomenon (the sequential effects of unintended/ unconsidered/ consequences) than what your studies explore (evaluation shifts after a single iteration of thinking about consequences).

• How do you actually operationalize consequences? How do you define and distinguish positive vs negative consequences? What exactly do you mean by “material but nonfocal consequences”? How do you decide which consequences are focal and which are not? These definitions need to be explicit and measurable. When you ask participants to generate "positive" and "negative" consequences, it is unclear how these are understood or differentiated. Are they interpreted as “benefits” vs “costs”? How do you ensure consistency across participants? As a result, your studies do not test or control for the instructions given for generating consequences. For example, what happens if participants are asked to generate only positive consequences or only negative ones? Or if you frame these as "benefits" and "costs"?

• While you claim that "negative consequences" are central to the phenomenon of consequence neglect (see e.g. your abstract), none of your studies actually examine whether it is specifically the negative (vs positive) consequences that drive the effect. You need to run additional study to test that. Additionally, neither of your studies tests the mechanism behind the “consequence neglect”.

• You mention "boundary conditions" for “consequence neglect” (on page 5), but this discussion is vague. What factors make consequences more or less focal and why? Provide clear predictions and test them.

• In the Introduction, you suggest that the paper focuses on policy designers, but your studies primarily involve general judgments about various social, business, health-related problems, without manipulating individual responsibility for implementing the policies/ solutions. In other words, real-world decision-makers typically focus on one domain (e.g., health, education) and they are incentivized to take responsibility for the outcomes. You did not manipulate this aspect. Instead, you test a wide variety of unrelated problems, which introduces some noise and ignores potential domain effects or the influence of prior expertise. For example, why was alcohol regulation significant in Study 2a but not in later studies? This should be explored. This gap weakens the practical implications of your work. Either tone down the policy implications or run additional studies in which you manipulate and test the effects of domain, context, and expertise.

• Neither of your study effectively address confounding factors, such as fatigue, participant expertise or fluency (which you admit on pages 27-29). Participants evaluated several diverse problems, which may have impacted their ability to focus.

• Your “Individual Difference Scale" (Study 1) lacks validation and it does not work anyway. Given this, along with the lack of control for confounds, I would consider dropping Study 1.

• I may be missing something but why didn’t you use repeated measures ANOVA with adjustments for multiple comparisons for your analyses?

Minor things:

• Please add an overview of all studies to clarify their purpose and connections.

• Please avoid unnecessary qualifiers like “compelling evidence” or “crucial ways.”

• Table 7 needs more details to aid interpretation.

In its current form, the paper is very promising but needs some revisions to strengthen the theory, clarify the constructs, and ensure that the findings are robust and relevant. Please rewrite the theory section to clearly position your work in relation to existing research. Consider removing your Study 1 and conducting a new one with better controls for fatigue, expertise, participant understanding of instructions etc. Explore alternative instructions for generating consequences to rule out fluency effects. How does the domain moderate the effects you observe? Consider focusing on testing real decision-makers or manipulating roles to better support your practical implications.

6. PLOS authors have the option to publish the peer review history of their article (what does this mean? ). If published, this will include your full peer review and any attached files.

**Do you want your identity to be public for this peer review?** For information about this choice, including consent withdrawal, please see our Privacy Policy .

Reviewer #1: No

Reviewer #2: No

---

## [Author Response · Author response to Decision Letter 1]

15 Mar 2025

Please see the response to the reviewers file for this submission.

---

## [Editor Report · Decision Letter 1]

17 Mar 2025

Side effects may include: Consequence neglect in generating solutions

PONE-D-24-48769R1

Dear Dr. Rodriguez,

We’re pleased to inform you that your manuscript has been judged scientifically suitable for publication and will be formally accepted for publication once it meets all outstanding technical requirements.

Kind regards,

Tobias Otterbring

Academic Editor

PLOS ONE

Additional Editor Comments (optional):

Dear authors,

Thanks for delivering a responsive revision. Based on your replies and edits in the manuscript, my assessment is that you have managed to compellingly counter most of the concerns raised by the reviewers. Therefore, I am happy to recommend acceptance of your piece for publication in PLOS One. Congratulations!

Kind regards,

Tobias Otterbring

Handling Editor, PLOS One
---

## [Editor Report · Acceptance letter]

PONE-D-24-48769R1

PLOS ONE

Dear Dr. Rodriguez,

I'm pleased to inform you that your manuscript has been deemed suitable for publication in PLOS ONE. Congratulations! Your manuscript is now being handed over to our production team.

Kind regards,

on behalf of

Professor Tobias Otterbring

Academic Editor

PLOS ONE